# Generation of the short TRIM32 isoform is regulated by Lys 247 acetylation and a PEST sequence

**Juncal Garcia-Garcia, Katrine Stange Overå, Waqas Khan, Eva Sjøttem** [ORCID]*

Department of Medical Biology, Autophagy Research Group, University of Tromsø–The Arctic University of Norway, Tromsø, Norway

* eva.sjottem@uit.no

**Data Availability Statement:** All relevant data are within the paper and its Supporting Information files.

**Funding:** The authors received no specific funding for this work.

## Abstract

TRIM32 is an E3 ligase implicated in diverse biological pathways and pathologies such as muscular dystrophy and cancer. TRIM32 are expressed both as full-length proteins, and as a truncated protein. The mechanisms for regulating these isoforms are poorly understood. Here we identify a PEST sequence in TRIM32 located in the unstructured region between the RING-BBox-CoiledCoil domains and the NHL repeats. The PEST sequence directs cleavage of TRIM32, generating a truncated protein similarly to the short isoform. We map three lysine residues that regulate PEST mediated cleavage and auto-ubiquitylation activity of TRIM32. Mimicking acetylation of lysine K247 completely inhibits TRIM32 cleavage, while the lysines K50 and K401 are implicated in auto-ubiquitylation activity. We show that the short isoform of TRIM32 is catalytic inactive, suggesting a dominant negative role. These findings uncover that TRIM32 is regulated by post-translational modifications of three lysine residues, and a conserved PEST sequence.

## Introduction

Tripartite motif (TRIM) proteins have emerged as a large family of ubiquitin E3 ligases that are involved in a wide range of cellular processes such as development, differentiation, immunity and carcinogenesis [1]. As a member of the TRIM family, TRIM32 shows the common RBCC domain organization in its N-terminal, consisting of a "Really interesting New gene" (RING) domain, followed by one B-box domain, and a Coiled Coil (CC) region [2]. TRIM32 is characterized by six NHL repeats in its C-terminus. The RING domain is essential for the E3 ligase activity, while the B-box domain is necessary to modulate chain assembly rate of ubiquitin units. The CC domain assumes an α-helix structure that allows the formation of anti-parallel dimers [3]. To date, the NHL repeats are found to be involved in dimerization and cargo recognition [4]. TRIM32 oligomerization is a pre-requisite for its catalytic activity.

TRIM32 has multiple target proteins involved in innate immunity, carcinogenesis and muscle physiology [5]. TRIM32 is linked to two different genetic diseases [4]. Mutations in the NHL domains result in the muscle disorders Limb Girdle Muscular Dystrophy 2H

**Competing interests:** The authors have declared that no competing interests exist.

(LGMD2H) and Sarcotubular myopathy (STM). These mutants display impaired oligomerization and auto-ubiquitylation activity, as well as reduced overall TRIM32 expression. A missense mutation in the B-box domain causes Bardet-Biedl Syndrome 11 (BBS11), which has a pleiotropic phenotype [6]. TRIM32 can be degraded both by the proteasome and by selective autophagy, and its auto-ubiquitylation activity is required for the autophagic degradation [7]. The autophagy receptor p62/SQSTM1 can direct ubiquitylated TRIM32 to autophagic degradation, but is at the same time a TRIM32 substrate, uncovering a role for TRIM32 in the regulation of selective autophagy.

Post-translational modifications (PTMs) such as phosphorylation, acetylation, ubiquitylation and SUMOylation are essential for the function and fate of most proteins. The attachment of these small molecules can modulate greatly the properties of a protein, affecting its stability, function, intracellular distribution and interaction with other proteins [8]. Acetylation is the most common PTM, playing important roles in cell signaling, and in the regulation of protein localization, stability and functionality. The addition of the acetyl group from an Acetyl coenzyme A (Ac-CoA) can occur in two different positions, in their $N^\alpha$-termini of the amino acid or in the $\varepsilon$-amino group of a lysine amino acid. Lysine acetylation is a reversible modification that is tightly controlled by multiple acetyltransferases and deacetylases [9]. This repertoire of addition and removal of molecules allow the cell to control fast and efficiently the different changes in the environment.

Ubiquitylation of proteins is implicated as a regulatory mechanism of many cellular processes. The addition of ubiquitin moieties is a reversible process that occur on lysine residues. Ubiquitin itself contains seven lysine residues that can be ubiquitylated, forming ubiquitin chains. This allows multiple combinations, which adds a high degree of complexity to this cellular regulation mechanism [10]. As acetylation, ubiquitylation processes are controlled by a balance between ubiquitin E3 ligases and de-ubiquitin enzymes.

A PEST sequence is a region within a protein that is rich in proline (P), glutamic acid (E), serine (S) and threonine (T) residues. This sequence of amino acids often acts as a proteolytic signal to control the rapid turnover of the protein [11]. Originally PEST sequences were attributed to short-lived proteins, but later PEST sequences are also identified in some long-lived proteins. Moreover, alternative functions have been linked to PEST domains, hence their function is not limited to proteolytic signaling [12]. PEST regions are generally unstructured and flexible, and the 26S Proteasome and other proteases such as calpains are in charge of the degradation of PEST containing proteins. PEST sequences can lead to constitutive degradation of the protein, but they can also behave as conditional degradation signals depending on the cellular needs [13].

In this work we identify a PEST sequence in the unstructured region between the RBCC region and the NHL domains of the ubiquitin E3 ligase TRIM32. The PEST sequence directs cleavage of TRIM32, leading to a truncated TRIM32 protein lacking the C-terminal NHL-repeats, resembling isoform 3 of TRIM32. The existence of TRIM32 isoform 3 is experimental evidenced at protein level, but a corresponding transcript is not verified. We uncover that PEST mediated cleavage of TRIM32 is regulated by post-translational modifications of three lysine residues, K50, K247 and K401. Mimicking acetylation on K247 completely protects TRIM32 from PEST mediated cleavage. K50 and K401 were found to be important for TRIM32 auto-ubiquitylation activity, which is reported to be necessary for its tetramerization and cytoplasmic body formation. This is the first time that a PEST sequence has been described in a TRIM protein, as well as the regulation of a PEST sequence by acetylation of an adjacent lysine residue.

## Results

### PTMs of the lysine residues K50, K247 and K401 regulate TRIM32 cleavage

It is well recognized that TRIM32 has auto-ubiquitylation activity in addition to conventional ubiquitin-protein ligase activity [14,15]. Auto-ubiquitylation seems to regulate its expression level, E3 ligase activity and ability to form cytoplasmic bodies [7,16–18]. We and others have shown that the missense mutation TRIM32$^{D487N}$ causing LGMD2H does not undergo auto-ubiquitylation [7,18]. Auto-ubiquitylated TRIM32 can be detected as a slower-migrating band on Western Blots, as indicated in Fig 1A where the slower migrating band is detected approximately 10 kDa above the band of the catalytic active EGFP-TRIM32$^{WT}$ and EGFP--TRIM32$^{P130S}$ [7]. The EGFP-TRIM32$^{D487N}$ mutant which do not contain auto-ubiquitylation activity [7], does not display this slow migrating band. Auto-ubiquitylation seems to be a prerequisite for the catalytic activity of TRIM32, as the LGMD2H mutant is unable to ubiquitylate the substrate protein p62/SQSTM1 [7]. Moreover, PKA mediated phosphorylation of TRIM32$^{S651}$, which impairs its ability to undergo auto-ubiquitylation, leads to repression of its E3 ligase activity. The PKA phosphorylation was shown to be regulated by 14-3-3 protein binding to soluble TRIM32, trapping it in a soluble and functionally latent complex [17]. In an attempt to better understand the regulation of TRIM32, we started out to identify which lysine residues in TRIM32 are targeted by its auto-ubiquitylation activity. For this purpose we applied Mass Spectroscopy (MS) analyses of the catalytic inactive disease mutant myc-TRIM32$^{D487N}$, and compared it to immunoprecipitated catalytic active myc-TRIM32$^{WT}$ and myc-TRIM32$^{P130S}$. All proteins were stably expressed in the HEK293 FlpIn TRIM32 knock out (KO) cells described previously [7]. No lysine residues of the catalytic inactive TRIM32$^{LGMD2H}$ disease mutant were detected as ubiquitylated, while peptides with K50 and K401 ubiquitylation were found in precipitates of the catalytic active enzymes myc-TRIM32$^{WT}$ and myc-TRIM32$^{P130S}$ (Fig 1B and 1C). However, peptides with acetylation of lysine K247 were identified in precipitates of the myc-TRIM32 disease mutants, but not in the TRIM32$^{WT}$ precipitates (Fig 1B and 1C). In order to determine if these lysine residues are targets for TRIM32 auto-ubiquitylation activity, the constructs EGFP-TRIM32$^{K50R}$, EGFP-TRIM32$^{K247R}$, and EGFP-TRIM32$^{K401R}$ were established by site-directed mutagenesis and transiently transfected into HEK293 FlpIn TRIM32 KO cells [7]. Western Blot analysis of the transfected proteins revealed that none of these single lysine to arginine mutations impaired auto-ubiquitylation activity of TRIM32 (S1A Fig). Next, we introduced double and triple K to R mutations in EGFP-TRIM32, generating EGFP-TRIM32$^{K247R/K401R}$, EGFP-TRIM32$^{K50R/K247R}$, EGFP-TRIM32$^{K50R/K401R}$ and EGFP-TRIM32$^{K50R/K247R/K401R}$, respectively. These mutants along with EGFP-TRIM32$^{WT}$, were transiently transfected into the TRIM32 KO cells and their auto-ubiquitylation activity investigated by Western blotting (Fig 1D). Surprisingly, introduction of the double mutation EGFP-TRIM32$^{K247R/K401R}$ resulted in a partially cleaved protein (Fig 1D and 1E, lane 3), while the triple EGFP-TRIM32$^{K50R/K247R/K401R}$ mutation resulted in a completely cleaved protein (Fig 1D and 1E, lane 6). Co-transfection of the broad specificity de-ubiquitinase USP2 verified that the slower-migrating band on the Western blots are due to ubiquitylated TRIM32 (Fig 1D and 1E and S1B Fig). Since mutations in TRIM32 is associated with muscular dystrophy, similar experiment was performed in the myoblast C2C12 cell line. Introduction of the K247R/K401R and K50/K247R/K401R mutations were exposed to cleavage also in this cell line (Fig 1E). Moreover, the polyclonal TRIM32 antibody displayed the same protein pattern on the Western blot as the GFP antibody, indicating that cleavage of EGFP-TRIM32$^{K50R/K247R/K401R}$ generates a stable N-terminal cleavage product of around 55 kDa. In contrast, any C-terminal cleavage products can not be detected in the gel, suggesting that this part of TRIM32 is exposed to further degradation.

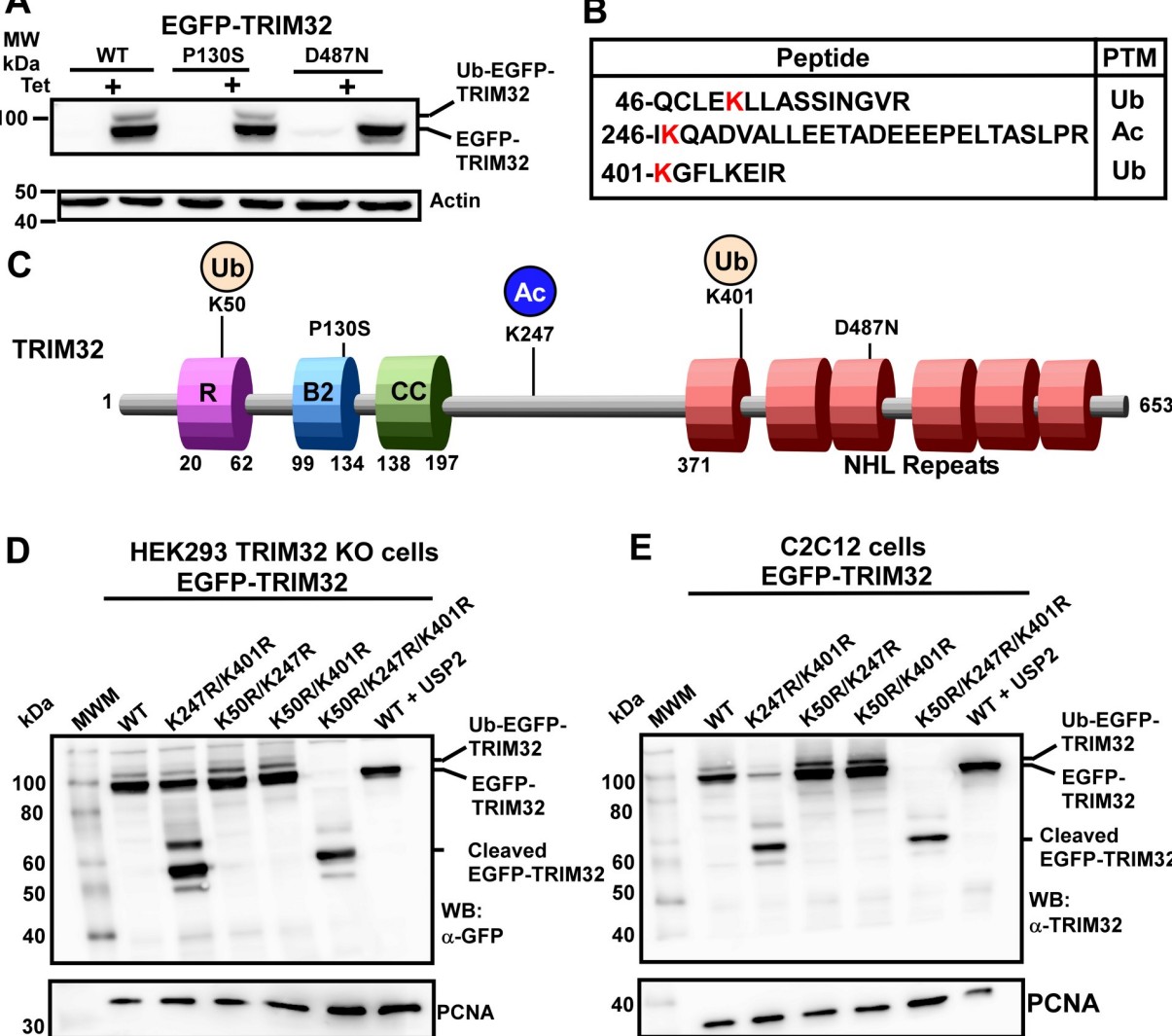

**Fig 1. PTMs of TRIM32^K50, TRIM32^K247 and TRIM32^K401 regulate its cleavage.** (A) Western blot analysis of cell extracts from HEK293 FlpIn cells with tetracycline inducible expression of EGFP-TRIM32^WT, the BBS11 associated mutant EGFP-TRIM32^P130S, or the LGMD2H causing mutant EGFP-TRIM32^D487N shows auto-ubiquitylation activity of TRIM32^WT and TRIM32^P130S, but not the LGMD2H mutant. (B) Sequence of the TRIM32 peptides identified by MS to display PTMs on lysine residues that differed between the TRIM32 proteins with auto-ubiquitylation activity (TRIM32^WT and TRIM32^P130S) and the LGMD2H disease variant lacking auto-ubiquitylation activity (TRIM32^D487N). (C) Schematic of TRIM32 domain organization. The D487N mutation in the NHL repeats that is associated with LGMD2H and the P130S mutation in the B-box causing BBS11 are indicated above. The three lysine residues that are focused in this paper are indicated in red, while the PTMs of these residues identified by MS are indicated above. Ub: Ubiquitylation. Ac: Acetylation (D) Western Blot analysis of HEK293 FlpIn TRIM32 KO cells transiently transfected with expression plasmids for EGFP-TRIM32^WT, the double mutants EGFP-TRIM32^K247R/K401R, EGFP-TRIM32^K50R/K247R, or EGFP-TRIM32^K247R/K401R, or the triple mutant EGFP-TRIM32^K50R/K247R/K401R. The last lane represents co-transfection of EGFP-TRIM32^WT and mCherry-USP2 expression vectors. The bands representing EGFP-TRIM32, auto-ubiquitylated EGFP-TRIM32, and cleaved EGFP-TRIM32 are indicated to the right. PCNA represents the loading control. (E) Western blot analysis of a similar experiment as in (D), except that the myoblast C2C12 cell line is used instead of the HEK293 FlpIn TRIM32 KO cell line, and the antibody used is an anti-TRIM32 antibody instead of an anti-GFP antibody.

## Acetylation of K247 is sufficient to inhibit cleavage of TRIM32

According to ENSEMBL genome browser (ensemble.org) and UniProt protein database (uniprot.org) there are two variants of the TRIM32 protein in humans. The main variant (Uniprot.org Q13049) is a 653 amino acid protein of 72 kDa. Additional, there is clear

experimental evidence for the existence of a short 172 amino acid long variant of TRIM32 (Uniprot.org Q5JVY0) (Fig 2A). However, the existence of a transcript encoding the short protein variant is not validated, since a transcript with a 3'UTR is unrecognized so far (ensemble. org). The short isoform contains the N-terminal RBCC domains, but lack the NHL-repeats and the unstructured region between RBCC and the NHL-domain. The short isoform is around 20 kDa, and hence has a size very similar to the 55 kDa EGFP-TRIM32 N-terminal cleavage product that we observed above (20 kDa plus the EGFP-tag of 32.7 kDa). This prompted us to ask whether the short TRIM32 protein may be generated by proteolytic cleavage of the long form, and if this cleavage could be regulated by post-translational modifications of the lysine residues identified above (K50, K247, K401)(Fig 1C). De-ubiquitylation of TRIM32 by USP2 over-expression did not result in TRIM32 cleavage (Fig 1D and 1E). Therefore, we asked whether acetylation of the lysine residues K50, K247 and K401 would affect TRIM32 cleavage. Each of these lysine residues were replaced by the acetylation mimicking amino acid glutamine (Q) in the EGFP-TRIM32$^{K50R/K247R/K401R}$ construct. The constructs were transfected into the HEK293 FlpIn TRIM32 KO cell line, and their expression monitored by Western blotting (Fig 2B). Clearly, introduction of the acetylation mimicking mutant at position K247 completely inhibited cleavage of TRIM32 (Fig 2B, lane 5), while the K50Q and K401Q mutations did not (Fig 2B, lanes 4 and 6). This suggests that acetylation of K247 in TRIM32 is implicated in regulation of TRIM32 cleavage. The subcellular localization of the TRIM32 lysine mimicking mutants were examined by transient transfection of wild type EGFP-TRIM32 and the mutant constructs into HEK293 FlpIn TRIM32 KO cells. In line with previous reports, EGFP-TRIM32$^{WT}$ is enriched in small round cytoplasmic bodies in addition to diffuse cytoplasmic localization. Co-staining with the Golgi marker GM130 showed that many of the TRIM32 bodies localize close to the Golgi apparatus (Fig 2C). The EGFP-TRIM32 mutants that generate the cleaved TRIM32 product (TRIM32$^{K50Q/k247R/K401R}$ and TRIM32$^{K50R/K247R/K401Q}$) formed mainly a few large bodies or aggregates localized in or close to the Golgi region. In contrast, the TRIM32$^{K50R/K247Q/K401R}$ mutant that inhibits TRIM32 cleavage, displays a subcellular localization very similar to the wild type protein (Fig 2C). Hence, the short TRIM32 protein seem to have a strong tendency to form large aggregates compared to full-length TRIM32 isoform.

## Autophagic degradation of TRIM32 is dependent on K50, K247 and K401 modifications

We have recently shown that TRIM32 is a substrate for selective autophagy [7]. Our next question was whether the short TRIM32 isoform is a target for selective autophagy. For this purpose, the double fluorescent tag mCherry-EYFP was cloned in front of the TRIM32 constructs TRIM32$^{WT}$, TRIM32$^{K247R/K401R}$, TRIM32$^{K247Q}$, TRIM32$^{K50R/K247Q/K401R}$, and TRIM32$^{K50R/K247R/K401R}$ and transiently transfected into the HEK293 FlpIn TRIM32 KO cells. Since EYFP is unstable in acidic environments while mCherry is stable, double-tagged TRIM32 constructs targeted to the lysosome will be visualized as RedOnly dots in the fluorescence microscope, while TRIM32 bodies in the cytoplasm will occur as yellow. The mCherry-EYFP-TRIM32$^{K247R/K401R}$ and mCherry-EYFP-TRIM32$^{K247Q}$ constructs formed RedOnly dots, but to a lesser extent than mCherry-EYFP-TRIM32$^{WT}$ (Fig 3). However, when all three lysine residues were substituted with Arginine, giving rise to the cleaved TRIM32 product, or when K50 and K401 were substituted with arginine and K247 with glutamine, giving rise to a full length TRIM32 product, no red only dots was observed in the transfected cells (Fig 3). This suggests that PTMs on the lysine residues K50, K247 and K401 regulate autophagic degradation of TRIM32.

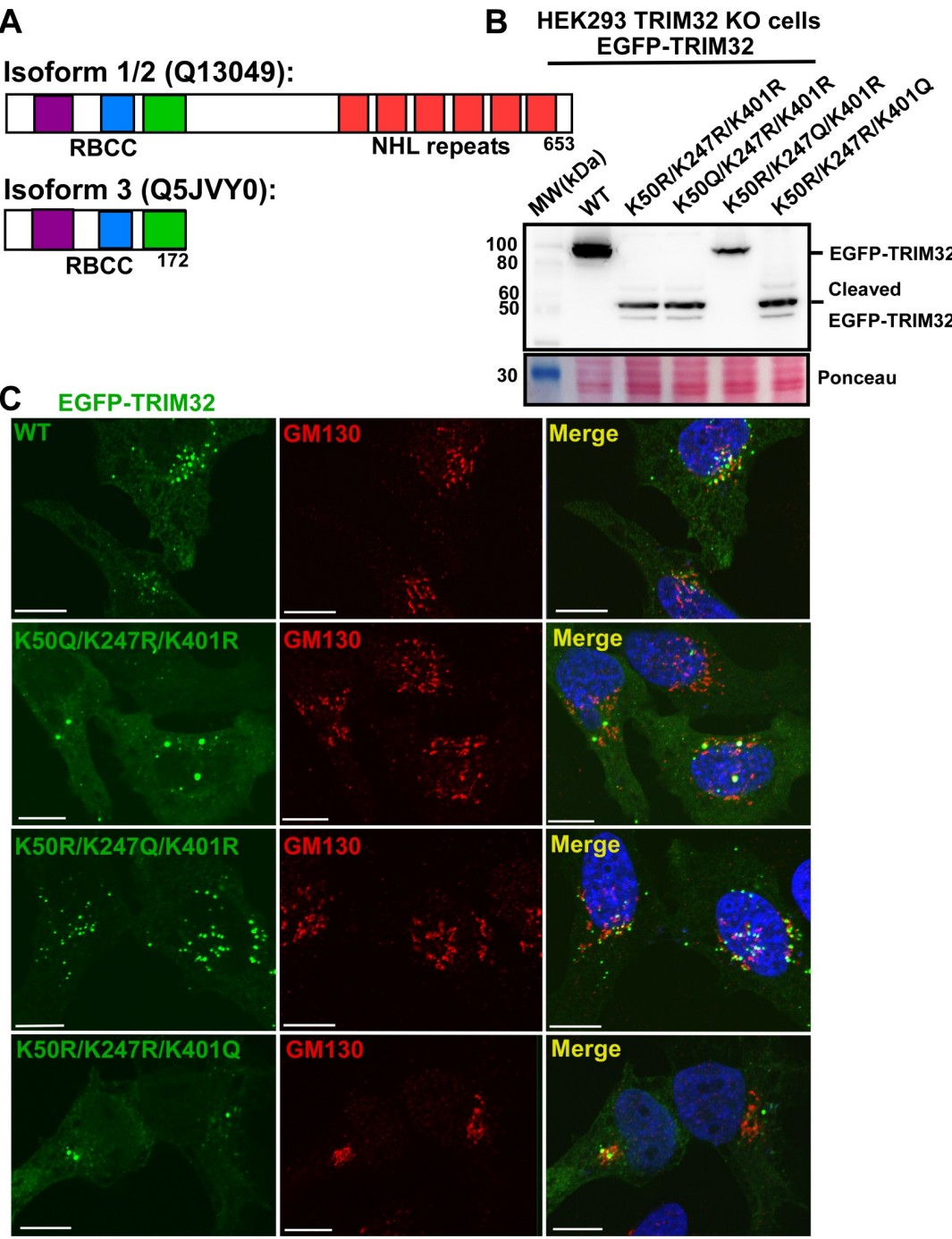

**Fig 2. K247 acetylation inhibits TRIM32 cleavage and facilitates its subcellular distribution.** (A) Schematic of the domain organization of TRIM32 isoform 1 and TRIM32 isoform 3. (B) Western Blot analysis of HEK293 FlpIn TRIM32 KO cells transiently transfected with expression plasmids for EGFP-TRIM32$^{WT}$, EGFP-TRIM32$^{K50R/K247R/K401R}$, EGFP-TRIM32$^{K50Q/K247R/K401R}$, TRIM32$^{K50R/K247Q/K401R}$ or EGFP-TRIM32$^{K50R/K247R/K401Q}$. The bands representing EGFP-TRIM32 and cleaved EGFP-TRIM32 are indicated to the right. Ponceau staining of the membrane represents the loading control. (C) Confocal images of HEK293 FlpIn TRIM32 KO cells transiently transfected with expression plasmids for EGFP-TRIM32$^{WT}$, EGFP-TRIM32$^{K50R/K247R/K401R}$, EGFP-TRIM32$^{K50Q/K247R/K401R}$, TRIM32$^{K50R/K247Q/K401R}$ or EGFP-TRIM32$^{K50R/K247R/K401Q}$, and immunostained with anti-GM130 antibodies, a Golgi marker. The cell nuclei are visualized by DAPI staining. Scale bars: 10 μM.

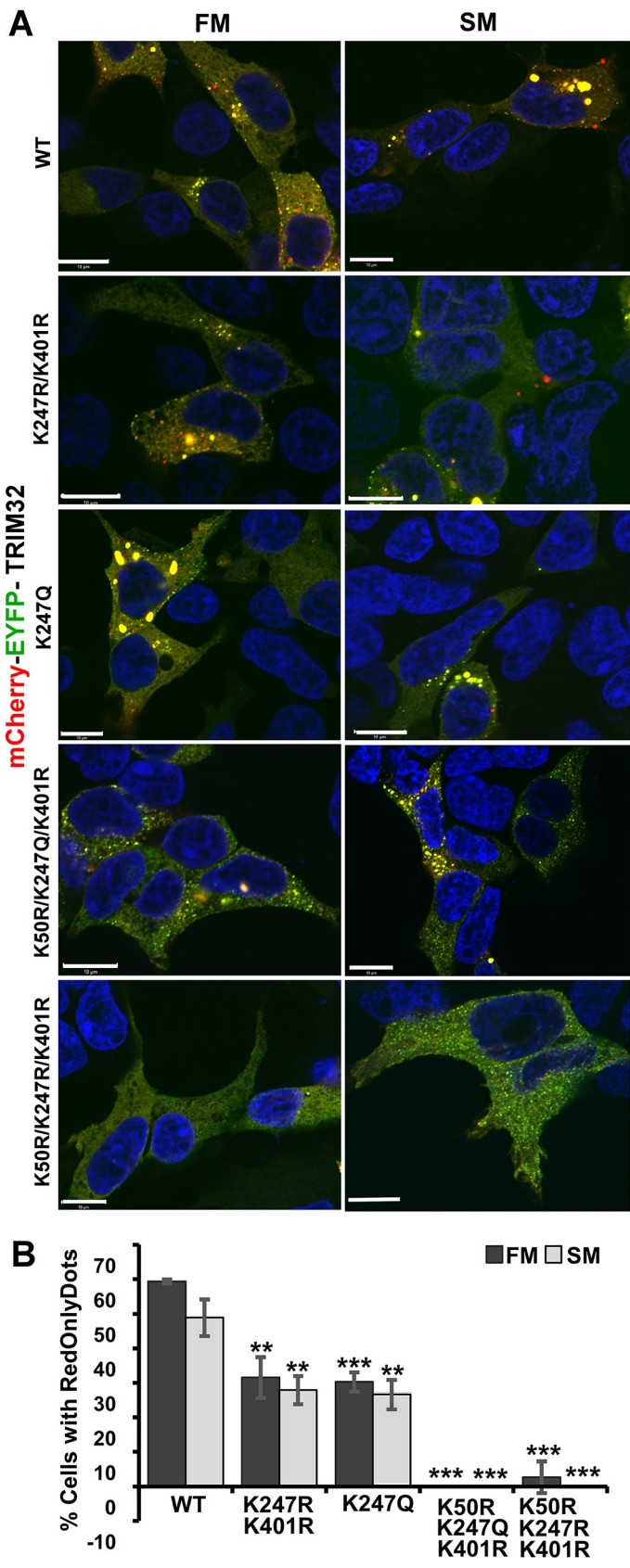

**Fig 3. PTMs on TRIM32$^{K50}$, TRIM32$^{K247}$, TRIM32$^{K401}$ regulate autophagic degradation of TRIM32.** (A) Confocal images of mCherry-EYFP-TRIM32$^{WT}$, mCherry-EYFP-TRIM32$^{K247R/K401R}$, mCherry-EYFP-TRIM32$^{K247Q}$, mCherry-EYFP-TRIM32$^{K50R/K247Q/K401R}$ or mCherry-EYFP-TRIM32$^{K50R/K247R/K401R}$ transiently expressed in HEK293 FlpIn TRIM32 KO cells and exposed to normal or starved conditions. Scale bars: 10 μM. (B) The graphs represent the number of cells with RedOnly dots in the mCherry-EYFP-TRIM32 transfected cells shown with representative images in A. The graphs represent the average of three independent experiments with s.d. (n>20 cells). $^{**}$P< 0.005; $^{***}$P<0.0005 (Student's *t*-test).

## TRIM32 cleavage is directed by a PEST sequence

The complete cleavage of the EGFP-TRIM32$^{K50R/K247R/K401R}$ construct prompted us to investigate whether TRIM32 encodes specific sequences that are exposed for proteolytic cleavage. Sequence analysis using the PEST prediction tool EMBOSS:epestfind (https://emboss.bioinformatics.nl/cgi-bin/emboss/epestfind), identified a putative PEST sequence with PEST score 7.4 located from amino acid 248 to 270 (Fig 4A). The predicted PEST sequence is 100% conserved in mammals, and interestingly it is located adjacent to lysine K247 which acetylation inhibits TRIM32 cleavage. To examine if the predicted PEST sequence directs TRIM32 cleavage, specific glutamate and threonine residues within the PEST sequence were mutated to valine or isoleucine, respectively, in the EGFP-TRIM32$^{WT}$ construct. Additionally, a partial deletion of the PEST sequence was introduced in EGFP-TRIM32$^{WT}$ and EGFP-TRIM32$^{K50R/K247R/K401R}$ (Fig 4A). The PEST mutation and deletion constructs were transiently transfected into the HEK293 FlpIn TRIM32 KO cell line, and their expression monitored by Western blotting (Fig 4B). Clearly, partial deletion of the PEST sequence in the EGFP-TRIM32$^{K50R/K247R/K401R}$ construct completely inhibits TRIM32 cleavage (Fig 4B, lane 6). Partial deletion or mutation of the PEST sequence in wild type TRIM32 did not compromise TRIM32 expression (Fig 4B, lanes 3 and 4). This identifies the 248–270 region of TRIM32 to be a PEST sequence that directs cleavage of TRIM32 that is unable to be modified on K50, K247 and K401. Clearly, mimicking of acetylation on lysine K247, which is localized adjacent to the PEST sequence, completely protects TRIM32 from the PEST directed cleavage (Fig 4B, last lane).

The cleaved TRIM32 protein contains the RBCC domains, and hence may have the ability to undergo auto-ubiquitylation and thereby gain catalytic activity. To examine this, the auto-ubiquitylation activity of the triple mutant TRIM32$^{K50R/K247R/K401R}$ was compared to the auto-ubiquitylation activity of TRIM32$^{WT}$ and TRIM32$^{D487N}$, which is known to be catalytic inactive. Clearly, no auto-ubiquitylation activity was detected for the cleaved TRIM32 product (Fig 4C, lane 5). Auto-ubiquitylation activity of TRIM32$^{K50R/K247R/K401R}$ with a partial deleted PEST sequence, and the acetylation mimicking mutant TRIM32$^{K50R/K247Q/K401R}$ were analyzed in parallel. Both these mutations inhibit cleavage of TRIM32 (Fig 4C, lanes 6,7). Notably, none of these mutant forms of TRIM32 displayed auto-ubiquitylation activity. In contrast, TRIM32$^{WT}$ with a partial deleted PEST sequence (Fig 4C, lane 4) and the TRIM32$^{K247Q}$ construct (Fig 4B, last lane) both display auto-ubiquitylation activity.

Together, these results show that the short isoform of TRIM32 can be generated by proteolytic cleavage directed by a PEST sequence located between the RBCC region and the NHL domains. Acetylation of K247 located adjacent to the PEST sequence, completely protects TRIM32 from PEST mediated cleavage. The short TRIM32 isoform and TRIM32 proteins containing K50, K247 and K401 mutations are catalytic inactive, showing the importance of these lysines for TRIM32 activity.

## Discussion

In this study we aimed at gaining insight in how TRIM32 auto-ubiquitylation regulates its activity and stability. TRIM32 is implicated in diverse biological and physiological processes

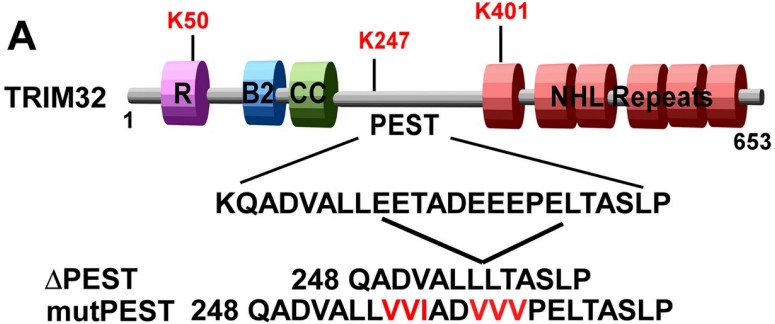

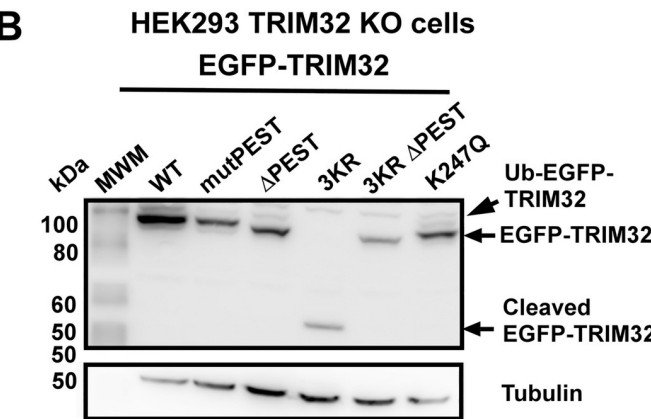

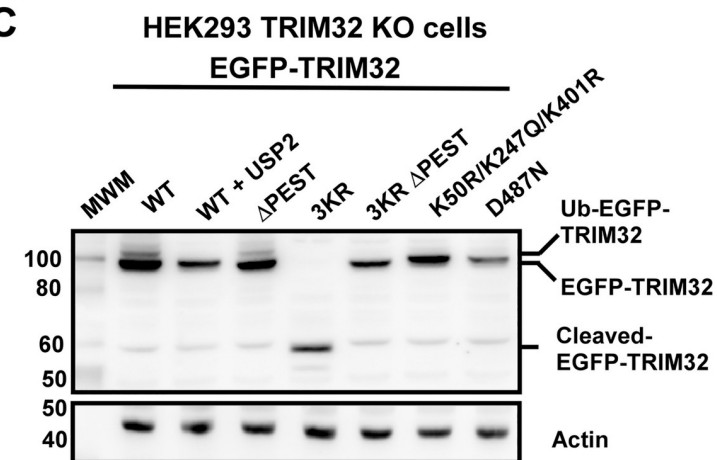

**Fig 4. Acetylation of K247 protects TRIM32 from cleavage directed by the adjacent PEST sequence.** (A) Schematic of TRIM32 domain organization with the three regulatory Lysine residues identified in this work indicated above, and the sequence of the mapped PEST sequence indicated below. MutPEST and ΔPEST display the modifications introduced to the predicted PEST sequence. (B) Western Blot analysis of cell extracts from HEK293 FlpIn TRIM32 KO cells transiently transfected with expression plasmids for EGFP-TRIM32$^{WT}$, EGFP-TRIM32$^{mutPEST}$, EGFP-TRIM32$^{ΔPEST}$, EGFP-TRIM32$^{K50R/K247R/K401R}$ (3KR), TRIM32$^{K50R/K247R/K401R/ΔPEST}$ (3KR ΔPEST) or EGFP-TRIM32$^{K247Q}$. The bands representing EGFP-TRIM32 and cleaved EGFP-TRIM32 are indicated to the right. Tubulin represents the loading control. (C) Western Blot analysis of cell extracts from HEK293 FlpIn TRIM32 KO cells transiently transfected with expression plasmids for EGFP-TRIM32$^{WT}$, EGFP-TRIM32$^{WT}$ and mCherry-USP2, EGFP-TRIM32$^{ΔPEST}$, EGFP-TRIM32$^{K50R/K247R/K401R}$ (3KR), TRIM32$^{K50R/K247R/K401R/ΔPEST}$ (3KR ΔPEST) or EGFP-TRIM32$^{K50R/247Q/K401R}$ or EGFP-TRIM32$^{D487N}$. The bands representing ubiquitylated EGFP-TRIM32, EGFP-TRIM32 and cleaved EGFP-TRIM32 are indicated to the right. Actin represents the loading control.

such as muscle physiology, neuronal differentiation, immunity and cancer [5]. TRIM32 is reported to act both as an oncogene and as tumor suppressor, depending on the specific organ and cellular context [5]. TRIM32 overexpression promotes cell proliferation, transforming activity, cell motility and prevents apoptosis [15,19], but it is also shown to promote asymmetric cell division of neuroblastoma cells and to enhance TNFα-induced apoptosis [20,21]. Abnormal expression of TRIM32 has been demonstrated in various human cancer cells [22–26], and in the occipital lobe of Alzheimer's disease patients [27]. Reduced TRIM32 expression in lung epithelial and tracheal cells increases their susceptibility to infection of influenza A virus [28]. Thus, regulating TRIM32 activity is of critical importance for healthy cell physiology.

Our data reveal that TRIM32 contains a conserved PEST sequence located in the unstructured region between the N-terminal RBCC domains and the C-terminal NHL-domains. This is the first time a PEST sequence is identified in a TRIM protein. Inhibition of PTMs of the K50, K247 and K401 residues by substitution of these lysine residues with arginine, resulted in exposure of the PEST to proteolytic enzymes, leading to TRIM32 cleavage. However, mimicking acetylation of K247 by glutamine substitution but keeping the K50R and K401R mutations, completely protected TRIM32 from proteolytic degradation. The acetylation also facilitated distribution of TRIM32 bodies throughout the cytoplasm, and protected it from autophagic degradation. Importantly, K247 is located immediately upstream of the PEST region. Thus, our data indicate that reversible acetylation of K247 regulates PEST mediated cleavage of TRIM32.

Research over the past decades have revealed that lysine acetylation is a common mechanism for regulation of protein stability, and can regulate both proteasome-dependent and lysosome-dependent protein degradation (recently reviewed [29]). Acetylation mediated stabilization of proteins can be due to direct competition between acetylation and ubiquitylation of the same lysine residue, such as described for SMAD7 [30] and TRIM50 [31]. However, acetylation is also reported to promote protein degradation by enhancing ubiquitylation leading to degradation in the proteasome [32,33] or the lysosome [34,35]. However, lysine acetylation regulating PEST mediated cleavage is to our knowledge not previously described, and hence presents a new mechanism for lysine acetylation mediating protein expression. Originally, PEST domains were identified in short living proteins [36], and later they are shown to function as an anchor site of E3 ubiquitin ligases required for ubiquitin dependent protein degradation [37,38]. TRIM32 degradation can be mediated both via proteasomal and lysosomal pathways [7]. Here we applied inhibitors of the proteasome, the lysosome and the calpain proteases (data not shown), but we were not able to pinpoint which pathway is implicated in the PEST mediated cleavage of TRIM32. We may speculate that the K247 acetylation inhibits binding of proteolytic enzymes, since acetylation of proteins is a well known mechanism for inhibiting (or promoting) protein-protein interactions [29].

TRIM32 activity is regulated by tetramerization via its coiled-coil domain [39]. Tetramerization leads to formation of RING domain dimers on each side of the two antiparallel TRIM32 dimers. RING domain dimerization is a common pre-requisite for E3 ligase activity of TRIM proteins [40,41]. TRIM32 tetramerization induces auto-ubiquitylation, which seems to be necessary for its conventional ubiquitin ligase activity and formation of cytoplasmic bodies. The scaffold 14-3-3 protein binds TRIM32 proteins phosphorylated at S651 in its very C-terminal end, and thereby inhibits TRIM32 tetramerization and cytoplasmic body formation [17]. HSP70, on the other hand, is reported to bind TRIM32 and promote formation of cytoplasmic bodies in an ATP-consuming process [42]. Apart from that, very little is known on how TRIM32 tetramerization and catalytic activity is regulated. The TRIM32[LGMD2H] disease mutants lack ubiquitylation activity and display a diffuse cytoplasmic localization [7,18]. This

is in line with the MS analysis of PTMs on the TRIM32$^{LGMD2H}$ mutant in this study, where no lysine residues were found to be ubiquitylated. However, lysine K247 in the mutant protein was found to be acetylated, indicating that TRIM32 acetylation is not dependent on tetramerization. Mutation of the lysine residues K50, K247 and K401 to arginine, lead to cleavage of EGFP-TRIM32 generating an around 55 kDa protein containing the N-terminal RBCC domains. This cleavage product resemble the short TRIM32 isoform, suggesting that isoform 3 of TRIM32 is generated by proteolytic cleavage. This finding is supported by the lack of a transcript that corresponds to isoform 3 (ensembl.org). Mimicking acetylation of lysine K247, but keeping the K50R and K401R mutations, protected TRIM32 from cleavage but did not restore its auto-ubiquitylation activity. This was in line with our MS results showing that the catalytic inactive TRIM32$^{LGMD2H}$ mutant can undergo K247 acetylation. Mimicking acetylation of K50 or K401 did not protect TRIM32 from PEST mediated cleavage and our MS data indicated that these residues are targets for ubiquitylation and not acetylation. Our results indicate that these two lysine residues are important for auto-ubiquitylation activity, and hence the E3 ligase activity of TRIM32.

Our study uncover that TRIM32 isoform 3 may be generated by PEST mediated cleavage of full-length TRIM32 isoform1/2, and that acetylation of lysine K247 localized adjacent to the PEST sequence protects TRIM32 isoform 1/2 from cleavage. We show that the short TRIM32 isoform is catalytic inactive, and hence may function as a dominant negative mutant regulating TRIM32 E3 ligase activity. Moreover, we find that auto-ubiquitylation activity of full-length TRIM32 is dependent on the lysine residues K50 and K401. K50 is located in the RING domain, and hence may directly affect the formation of a functional catalytic unit. K401 is located in the NHL repeats. Various mutations in the NHL repeats cause LGMD2H, and we have previously shown that TRIM32$^{LGMD2H}$ mutants are catalytic inactive [7].

Identification of a functional PEST sequence in a TRIM family protein is a novel finding, and also that acetylation of a lysine residue adjacent to the PEST region regulates the exposure of the PEST. These findings may contribute to the understanding of cellular mechanisms leading to dysregulated TRIM32 expression observed in pathological conditions.

## Materials and methods

### Antibodies

The following primary antibodies were used: rabbit polyclonal antibody for TRIM32 (Proteintech, 10326-1-AP); rabbit polyclonal anti-GFP (Abcam, ab290); rabbit polyclonal anti-Actin (Sigma, A2066); mouse monoclonal anti-PCNA (DAKO, M0879); rabbit monoclonal anti-GM130 (Abcam, #52649). The following secondary antibodies were used: Horseradish-peroxidase (HRP)-conjugated goat anti-rabbit IgG (BD 5 Biosciences, 554021); HRP-conjugated goat anti-mouse Ig (BD Biosciences, 554002); HRPconjugated anti-Biotin antibody (Cell Signalling, #7075), and Alexa FluorR 555-conjugated goat anti-rabbit IgG (Life Technologies, A-11008).

### Cell culture and transfections

HEK293 FlpIn T-Rex (ThermoFisher, R714-07), HEK293 FlpIn T-Rex TRIM32 KO [7], C2C12 (ATCC$^{®}$ CRL-1772$^{™}$) were cultured in Dulbecco's modified eagle's medium (DMEM) (Sigma, D6046) with 10% fetal bovine serum and 1% streptomycin-penicillin (Sigma, P4333). Sub-confluent cells in 6-well disheswere transfected using Metafectene Pro (Biontex, T040) following the manufacturer's instructions.

## Plasmids

All plasmids used in this study are listed in Table 1. Plasmids were made by conventional restriction enzyme based cloning or by use of the Gateway recombination system (Thermo-Fisher). Gateway LR reactions were performed as described in the instruction manual. Point mutations and deletion were carried out using the Site-directed-mutagenesis kit from STRA-TAGENE, using the primers described in Table 2. The oligonucleotides were ordered from ThermoFisher. All plasmids were verified by DNA sequencing (BigDye, Applied Biosystems, 4337455).

## Western blotting

Cells were seeded in 6-well dishes and transfected as indicated. One day post transfection the cells were lysed in 1xSDS buffer (50mM Tris pH 7.4; 2% SDS; 10% Glycerol) supplemented with 200mM dithiothreitol (Sigma, #D0632) and heated at 100°C for 10 minutes. The lysates were resolved by SDS-PAGE and transferred to nitrocellulose membrane (Sigma, GE10600003). The membrane was stained with Ponceau S (Sigma, P3504), blocked with 5% non-fat dry milk in 1% TBS-T (0.2M Tris pH 8; 1.5M NaCl and 0.05% Tween20 (Sigma, P9416)) and then incubated with indicated primary antibodies for 24h. The membrane was washed three times for 10 minutes each with PBS-T followed by incubation with secondary antibody for 1h. The membrane was washed three times for 10 minutes and analyzed by enhanced chemiluminescence using the ImageQuant LAS 4000 (GE Lifescience).

**Table 1. Plasmids used in this study.**

| | |
|---|---|
| pDest EGFP-TRIM32 | [7] |
| pDest EGFP-TRIM32$^{D487N}$ | [7] |
| pDest EGFP-TRIM32$^{P130S}$ | [7] |
| pDest EGFP-TRIM32$^{K50R}$ | This study |
| pDest EGFP-TRIM32$^{K247R}$ | This study |
| pDest EGFP-TRIM32$^{K401R}$ | This study |
| pDest EGFP-TRIM32$^{K247R/K401R}$ | This study |
| pDest EGFP TRIM32$^{K50R/K247R}$ | This study |
| pDest EGFP-TRIM32$^{K50Q/K401R}$ | This study |
| pDest EGFP-TRIM32$^{K50R/K247R/K401R}$ | This study |
| pDest EGFP-TRIM32$^{K50Q/K247R/K401R}$ | This study |
| pDest EGFP-TRIM32$^{K50R/K247Q/K401R}$ | This study |
| pDest EGFP-TRIM32$^{K50R/K247R/K401Q}$ | This study |
| pDest EGFP-TRIM32$^{\Delta PEST}$ | This study |
| pDest EGFP-TRIM32 $^{K50R/K247R/K401R\ \Delta PEST}$ | This study |
| pDest EGFP-TRIM32$^{mutPEST}$ | This study |
| pDest EGFP-TRIM32$^{K247Q}$ | This study |
| pDest mCherry-USP2 | This study |
| pDest mCherry-EYFP-TRIM32 | [7] |
| pDest mCherry-EYFP-TRIM32$^{K247R/K401R}$ | This study |
| pDest mCherry-EYFP-TRIM32$^{K247Q}$ | This study |
| PDest mCherry-EYFP-TRIM32$^{K50R/K247Q/K401R}$ | This study |
| pDEST mCherry-EYFP-TRIM32$^{K50R/K247R/K401R}$ | This study |
| pDest EGFP-C1 | [43] |
| pDest mCherry-EYFP | [44] |

**Table 2. Oligonucleotides used in this study.**

| | |
|---|---|
| TRIM32$^{K50R}$ | 5'-TGCCGCCAGTGCCTGGAGCGCCTATTGGCCAGTAGCATC- 3' |
| TRIM32$^{K247R}$ | 5'-TACTTCCTGGCCAAGATCCGCCAGGCAGATGTAGCACTA- 3' |
| TRIM32$^{K401R}$ | 5'-ATACAAGTCTTTACCCGCCGCGGCTTTTTGAAGGAAATC- 3' |
| TRIM32$^{K50Q}$ | 5'-TGCCGCCAGTGCCTGGAGCAGCTATTGGCCAGTAGCATC- 3' |
| TRIM32$^{K247Q}$ | 5'-TACTTCCTGGCCAAGATCCAGCAGGCAGATGTAGCACTA- 3' |
| TRIM32$^{K401Q}$ | 5'-ATACAAGTCTTTACCCGCCAAGGCTTTTTGAAGGAAATC- 3' |
| TRIM32$^{\Delta PEST}$ | 5'-AGGCAGATGTAGCACTACTGCTCACTGCCAGCTTGCCTCG- 3' |
| TRIM32$^{mutPEST}$ | 5'- CACTACTGGTGGTGATAGCTGATGTGGTGGTGCCAGAGCT- 3' |

## Immunostaining

Subconfluent cells were grown on coverslips (VWR, #631–0150) coated with Fibronectin (Sigma, F1141). They were fixed in 4% formaldehyde for 20min at R.T., permeabilized with methanol at RT for 5min, blocked in 5% goat serum/PBS or 5% BSA/PBS and incubated at room temperature with a specific primary antibody followed by Alexa Fluor 555 conjugated secondary antibody and DAPI. Confocal images were obtained using a 63x/NA1.4 oil immersion objective on an LSM780 system and the ZEN software (Zeiss). Quantification of cells containing red only dots in the mCherry- EYFP-double tag assay was done manually in three independent experiments, each including at least 30 cells.

## Statistics

All experiments were repeated at least three times, unless otherwise specified. Error bars represent the standard deviations. Two-sided unpaired, homoscedastic Student T-Tests were performed to assess significant differences between populations. Replicates were not pooled for statistical analyses.

## Supporting information

**S1 Fig. Inhibition of proteolytic enzymes does not stabilize the TRIM32$^{K50R/K247R/K401R}$ mutant or the TRIM32$^{K247R/K401R}$ mutant. A.** Western Blot analysis of HEK293 FlpIn TRIM32 KO cells transiently transfected with expression plasmids for EGFP-TRIM32$^{WT}$, EGFP-TRIM32$^{K50R}$, EGFP-TRIM32$^{K247R}$, or TRIM32$^{K401R}$. The bands representing EGFP-TRIM32 and auto-ubiquitylated EGFP-TRIM32 are indicated to the right. PCNA represents the loading control. **B.** Western Blot analysis of cell extracts from HEK293 FlpIn TRIM32 KO cells transiently transfected with expression plasmids for EGFP-TRIM32$^{WT}$, EGFP-TRIM32$^{K50R/K401R}$, EGFP-TRIM32$^{K50R/K247R}$ or EGFP-TRIM32$^{K247R/K401R}$. Expression plasmid for mCherry-USP2 is co-transfected where indicated. The bands representing EGFP-TRIM32, auto-ubiquitylated EGFP-TRIM32 and cleaved EGFP-TRIM32 are indicated to the right Ponceau represents the loading control.
(PDF)

**S1 Raw images.**
(PDF)

## Acknowledgments

We thank the Tromsø University Proteomics Platform for help with mass spectrometry analysis and the Advanced Microscopy Core Facility, UiT-The Arctic University of Norway, for the use of instrumentation.

## Author Contributions

**Conceptualization:** Juncal Garcia-Garcia, Eva Sjøttem.

**Formal analysis:** Juncal Garcia-Garcia, Eva Sjøttem.

**Investigation:** Katrine Stange Overå, Waqas Khan, Eva Sjøttem.

**Methodology:** Juncal Garcia-Garcia, Katrine Stange Overå, Waqas Khan, Eva Sjøttem.

**Project administration:** Eva Sjøttem.

**Supervision:** Eva Sjøttem.

**Writing – original draft:** Juncal Garcia-Garcia, Eva Sjøttem.

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
