## [Decision Letter · Decision Letter 0]

12 Feb 2021

PONE-D-21-00733

Generation of the short TRIM32 isoform is regulated by Lys 247 acetylation and a PEST sequence

PLOS ONE

Dear Dr. Sjøttem,

Thank you for submitting your manuscript to PLOS ONE. This manuscript was reviewed by an external reviewer and myself (AE of PLOS One).  Our reviews are similar though that of mine is bit more optimistic. However we both address very similar concerns that should be addressed. One of our main concerns is that there are very strong conclusions that are not clearly supported by experiments or the figures. After careful consideration, we feel that it has merit but does not fully meet PLOS ONE’s publication criteria as it currently stands. Therefore, we invite you to submit a revised version of the manuscript that addresses the points raised during the review process.

We look forward to receiving your revised manuscript.

Kind regards,

Michael Massiah

Academic Editor

PLOS ONE

Journal Requirements:

Additional Editor Comments:

Review from AE

I have reviewed your manuscript. It worthy of publication however, there are a few comments that I have that should be straightforward to address.

Can Fig 1A be bigger because as shown, the separation of the mono-Ub TRIM32 and TRIM32 seems too small to be the molecular weight of Ub?Am I to assume that TRIM32 is only capable of mono-ubiquitination?I am a bit confused that fig1 does not clearly show TRIM32 20K isoform or its 20K cleavage product. It appears that the band is at 40K. Also, it appears that TRIM32 (being 70k) is migrating at over 100K, can these be clarified?The statement (ln185) stating that acetylation of K247 is implicated in TRIM32 cleavage seems a bit strong because the structural implication of the K247 mutation is not clear.Lines 190-194 suggest that some of these mutant TRIM32 constructs form clump near the golgi? ER/Golgi produced proteins are usually destined for membrane association or extracellular. Is TRIM32 such a protein? Also the fact that these proteins form clumps suggests that the mutations may cause protein unfolding. Is there any validation that this is indeed the case or not?The conclusions of the PEST studies are based on fig 4b. The figure is too faint to make such conclusion, at least convincingly to this reader. In addition, the PEST sequence identified does not seem to have the signature P-S-T residues. Deletion of residues of this region could have structural implications. Can you clarify this some more.

Last, the manuscript would benefit from a careful read as there are a few minor grammar/typo issues

Reviewers' comments:

Reviewer's Responses to Questions

**Comments to the Author**

1. Is the manuscript technically sound, and do the data support the conclusions?

Reviewer #1: No

2. Has the statistical analysis been performed appropriately and rigorously? 

Reviewer #1: Yes

3. Have the authors made all data underlying the findings in their manuscript fully available?

Reviewer #1: Yes

4. Is the manuscript presented in an intelligible fashion and written in standard English?

Reviewer #1: No

5. Review Comments to the Author

Reviewer #1: The manuscript entitled "Generation of the short TRIM32 isoform is regulated by Lys 247 acetylation and a

PEST sequence" submitted by Sjottem et. all had concluded the following:

Authors suggest that three lysine residues regulates PEST mediating cleavage

and auto-ubiquitylation activity of TRIM32, Further authors also suggest that lysine K247 completely inhibits TRM32 cleavage. Lysines K50 and K401 regulate autoubiquitylation activity.

There are several lack of experimental proof before this manuscript can be considered for the publication.

There is no data showing the lack of autoubiquitination of TRIM32. Further the modification of Lysine (K) 247 is essential for cleavage. Authors have not shown any data if this mutation also effect its ubiquitination. Similairly no biochemical evidence had been provided for the Lysines K50 and K401 regulated autoubiquitylation activity.

Authors have not shown the rescue of the cleavage either inhibiting ubiquitin proteasome system or autophagy.

Does PEST sequence only important for the the cleavage or its modification effect its cleavage or functional autophagy.

Authors need to work further before it can be further considered for publication.

6. PLOS authors have the option to publish the peer review history of their article (what does this mean?). If published, this will include your full peer review and any attached files.

Reviewer #1: No

---

## [Author Response · Author response to Decision Letter 0]

17 Feb 2021

Dear Reviewers,

Please find enclosed the revised manuscript by Garcia-Garcia et al. entitled “Generation of the short TRIM32 isoform is regulated by Lys 247 acetylation and a PEST sequence”. We are grateful for the opportunity to submit a revised version of the manuscript, and thank the editor and the reviews for their useful comments to improve the paper. We have revised the manuscript according to reviews suggestions and criticisms as listed below.

The following is changed in the revised manuscript (MS): 

1. The Figures are scaled to fit within maximum sizes according to the author guidelines

2. All changes in the text according to the reviews comments are marked by red letters.

3. We provide Supporting Information showing the original uncropped and unadjusted images underlying all Western blots results reported in Figures and Supplemental figures.

Below is our response to the comments made by the editor and the reviewers:

Review from AE

I have reviewed your manuscript. It worthy of publication however, there are a few comments that I have that should be straightforward to address.

1. Can Fig 1A be bigger because as shown, the separation of the mono-Ub TRIM32 and TRIM32 seems too small to be the molecular weight of Ub?

Answer: Fig. 1A is made bigger in the revised version. Moreover, we have clarified in the text that it is shown previously by us (Overå et al., 2019) and others (Locke et al., 2009) that the slower migrating band (migrates as an around 10 kDa larger protein) is due to auto-ubiquitylation (new text marked with red, page 4/5). It is also shown previously that TRIM32D487N is unable to undergo auto-ubiquitylation. In. Fig1A we clearly see that this mutant form of TRIM32 does not display a slower-migration band. We also show in Supplemental Fig. 1B, and and in the last lanes of Fig. 1D and 1E, that this slower migrating band disapperars when over-expression of the de-ubiquitinase USP2. This indicated that the slower-migrating band is due to mono-ubiquitylation. This is mentioned in the results section first part of page 6).

2. Am I to assume that TRIM32 is only capable of mono-ubiquitination?

Answer: TRIM32, as many other RING E3 ligases, is able both to mono and polyubiquitylate other proteins. This has been published for TRIM32 (see f.ex. Locke et al., 2009). However, the self – or – auto-ubiquitylation seems to be mainly mono-ubiquitylation. This mono-ubiquitylation activity comes from oligomerization of TRIM E3 ligases, which is e pre-requisite for their catalaytic activity (Koliopoulos et al., 2016). When TRIM32 is highly expressed, more slower migrating bands above the mon-Ub band can be detected, but they are much weaker than the mono-Ub band (see ex. Overå et al., 2019).

3. I am a bit confused that fig1 does not clearly show TRIM32 20K isoform or its 20K cleavage product. It appears that the band is at 40K. Also, it appears that TRIM32 (being 70k) is migrating at over 100K, can these be clarified?

Answer: We are sorry that we have not made this more clearly. In all experiments we have used HEK293 FlpIn cells which are knocked out for TRIM32 (see establishment of the cell line in Overå et al., 2019). TRIM32 WT and mutants have been introduced to this cell line as EGFP-fusion proteins. Hence, you have to add the size of EGFP (32.7 kDa) to the size of TRIM32 or the size of the cleavage product, to find the correct bands on the gel. Thus full length TRIM32 will run at around 100 kDa, while the cleaved product will be between 50 and 60 kDa. We have indicated above the WB gels that it is EGFP-TRIM32, and in the revised Figures we have marked to the right of each blot where the full length EGFP-TRIM32 and the cleaved EGFP-TRIM32 product are seen. We have also explained this better in the result section of the revised manuscript.

4. The statement (ln185) stating that acetylation of K247 is implicated in TRIM32 cleavage seems a bit strong because the structural implication of the K247 mutation is not clear.

Answer: We do show in Supplemental Figure 1A, that mutation of K247 to arginine (K247R) does not affect the migration or the auto-ubiquitylation of TRIM32, suggesting that the structure of the protein is not rigoursly affected. However, mimicking acetylation of K247 by the K to Q mutation may locally change the structure of the protein. This local change would be similar to the change that an acetylation of K247 leads to – and in our MS data we identified that this residue is able to be acetylated. Hence, we state that acetylation of this residue protects TRIM32 from cleavage. The protection can be mediated through structural changes that inhibits exposure of the adjacent PEST sequence, or through a local change in the electrostatical interaction with other proteins.

Lines 190-194 suggest that some of these mutant TRIM32 constructs form clump near the golgi? ER/Golgi produced proteins are usually destined for membrane association or extracellular. Is TRIM32 such a protein? Also the fact that these proteins form clumps suggests that the mutations may cause protein unfolding. Is there any validation that this is indeed the case or not?

Answer: TRIM32 is mainly not a membrane associated protein. It is diffuse in the cytoplasm, in addition to forming cytoplasmic bodies. This has been published by us and others previously (Locke et al., 2009; Overå et al., 2019). In Figure 2 we show that introduction of three lysine mutations, K50R, K247R and K401R, leads to mis-distribution of TRIM32 in the cytoplasm. This can be due to a more unstable folding of the short TRIM32 isoform, leading to more aggregation. We also show that mimicking acetylation on K247 rescue the distribution pattern of TRIM32. 

5. The conclusions of the PEST studies are based on fig 4b. The figure is too faint to make such conclusion, at least convincingly to this reader. In addition, the PEST sequence identified does not seem to have the signature P-S-T residues. Deletion of residues of this region could have structural implications. Can you clarify this some more.

Answer: We have substituted the Western blot in Fig. 4B with a new blot showing stronger bands. Also Fig. 4C shows the cleavage of TRIM32 when the three lysine residus K50, K247 and K401 are mutated to arginine (lane 5), but when parts of the PEST sequence is deleted in this mutated TRIM32 protein, TRIM32 is not cleaved anymore and migrates as the wild-type protein. Lanes 4 of Fig. 4 B and C show that deletion of the PEST does not inhibit TRIM32 auto-ubiquitylation, suggesting that the structure of the protein is functional. The PEST sequence was predicted by the bioinformatics tool EMBOSS:epestfind. The link is included in the revised manuscript. A PEST sequence is rich in proline (P), glutamic acid (E), serine (S), and threonine (T) (Rogers et al., 1986) – and the TRIM32(248-270) region is rich in these amino-acids. 

Last, the manuscript would benefit from a careful read as there are a few minor grammar/typo issues.

Answer: We have read carefully through the manuscript and corrected typo issues.

Reviewers' comments:

Reviewer #1: The manuscript entitled "Generation of the short TRIM32 isoform is regulated by Lys 247 acetylation and a

PEST sequence" submitted by Sjottem et. all had concluded the following:

Authors suggest that three lysine residues regulates PEST mediating cleavage

and auto-ubiquitylation activity of TRIM32, Further authors also suggest that lysine K247 completely inhibits TRM32 cleavage. Lysines K50 and K401 regulate autoubiquitylation activity.

There are several lack of experimental proof before this manuscript can be considered for the publication.

There is no data showing the lack of autoubiquitination of TRIM32. 

Answer: In Fig. 1A, last lane, we show the lack of auto-ubiquitylation of TRIM32 associated with the LGMD2H disease mutation. We also show in Fig. 1D and Fig. 1E, that over-expression of the de-ubiquitinase USP2 inhibits TRIM32 auto-ubiquitylation. 

Further the modification of Lysine (K) 247 is essential for cleavage. Authors have not shown any data if this mutation also effect its ubiquitination. 

Answer: In Supplemental Fig. 1A, we show that the K247R mutation do not affect auto-ubiquitylation of TRIM32. Also, in Fig. 4B, last lane, we see auto-ubiquitylation of the K247Q mutated TRIM32. Hence, mutation of K247 does not seem to inhibit auto-ubiquitylation. In the revised manuscript, we have pinpointed to specific lanes on the blots that show auto-ubiquitylation and cleavage. 

Similairly no biochemical evidence had been provided for the Lysines K50 and K401 regulated autoubiquitylation activity.

Answer: In Fig. 4C, lane 7, we show that auto-ubiquitylation of TRIM32 is inhibited when the K49R and K40R mutations are introduced into the TRIM32 K247Q construct. In contrast, introduction of K247Q mutation into TRIM32 does not inhibit auto-ubiquitylation (last lane Fig. 4B). This suggests that K49 and K401 are important for auto-ubiquitylation activity.

To clarify this more for the reader, the result section page 10 is partial rewritten in the revised manuscript.

Authors have not shown the rescue of the cleavage either inhibiting ubiquitin proteasome system or autophagy.

Answer: As stated in the discussion (lines 316 – 318), we were not able to inhibit cleavage by adding the proteasome inhibitor MG132, or by adding the lysosomal inhibitor Bafilomycin A1, to the cells. Adding both inhibitors to the cells were very toxic for the cells, hence we did not get any conclusive results. In the autophagy assay (Figure 3) we show that mutation of the three lysine K49, K247 and K401 inhibits autophagic degradation. 

Does PEST sequence only important for the the cleavage or its modification effect its cleavage or functional autophagy.

Answer: The focus of this work was to show that the PEST sequence is important for cleavage. Fig. 4 shows that TRIM32 with mutated or partial deleted PEST sequence contains auto-ubiquitylation activity. This suggest that the PEST is not important for the E3 ligase activity of TRIM32.

---

## [Editor Report · Decision Letter 1]

23 Apr 2021

Generation of the short TRIM32 isoform is regulated by Lys 247 acetylation and a PEST sequence

PONE-D-21-00733R1

Dear Dr. Sjøttem,

We’re pleased to inform you that your manuscript has been judged scientifically suitable for publication and will be formally accepted for publication once it meets all outstanding technical requirements.

Kind regards,

Michael Massiah

Academic Editor

PLOS ONE
---

## [Editor Report · Acceptance letter]

7 May 2021

PONE-D-21-00733R1 

Generation of the short TRIM32 isoform is regulated by Lys 247 acetylation and a PEST sequence 

Dear Dr. Sjøttem:

I'm pleased to inform you that your manuscript has been deemed suitable for publication in PLOS ONE. Congratulations! Your manuscript is now with our production department. 

Kind regards, 

on behalf of

Dr. Michael Massiah 

Academic Editor

PLOS ONE